# Experimental Study: Interleukin-31 Augments Morphine-Induced Antinociceptive Activity and Suppress Tolerance Development in Mice

**DOI:** 10.3390/ijms242216548

**Published:** 2023-11-20

**Authors:** Iwao Arai, Minoru Tsuji, Saburo Saito, Hiroshi Takeda

**Affiliations:** 1Department of Pharmacology, International University of Health and Welfare, 2600-1 Kitakanemaru, Ohtawara 324-8510, Japan; 2Division of Environmental Allergy, The Jikei University School of Medicine, 3-25-8 Nishi-Shinbashi, Tokyo 105-8461, Japan

**Keywords:** analgesia, antinociception, interleukin-31 (IL-31), interleukin receptor A (IL-31RA), IL-31 receptor A-deficient (IL-31RAKI) mice, itch, morphine, pain

## Abstract

Morphine-induced antinociception is partially reduced in interleukin-31 (IL-31) receptor A (IL-31RA)-deficient mice, indicating that IL-31RA is crucial for morphine-induced peripheral antinociception. Herein, we examined the combined effects of IL-31 and morphine on the antinociceptive activity and itch-associated scratching behavior (LLS) in mice and elucidated the regulatory mechanisms. A hot-plate test was used to assess antinociception. LLS was automatically detected and recorded via a computer. IL-31RA mRNA expression was assessed using real-time polymerase chain reaction. Repeated pre-treatment with IL-31 resulted in significant antinociceptive activity. Repeated administration of morphine decreased the morphine-induced antinociceptive activity, LLS counts, and regular dose and inhibited IL-31-induced LLS. These results suggested that the repeated administration of morphine depleted inter-neuronal IL-31RA levels, preventing morphine-induced antinociception. Therefore, IL-31 may be helpful as an adjunct analgesic to morphine. To explore the benefits of IL-31, its influence on morphine-induced antinociceptive tolerance in mice was examined. An IL-31 and morphine combination increased the analgesic action, which increased the expression of DRG neuronal IL-31RA, elucidating the site of peripheral antinociception of morphine. This site may induce exocytosis of IL-31RA in the sensory nervous system. Collectively, the suppressive effect of IL-31 on morphine-induced antinociceptive tolerance may result from IL-31RA supplementation in sensory nerves.

## 1. Introduction

Opioids (such as morphine) are the most effective drugs for treating severe pain. However, these treatments are accompanied by several adverse events, including itching, tolerance, dependence, nausea, constipation, sedation, and respiratory depression. Furthermore, the analgesic efficacy of opioids varies among individuals [1,2]. Therefore, effective pain treatment is often hampered by considerable differences in opioid sensitivity. Insufficient opioid doses can lead to inadequate pain relief, whereas unnecessarily high doses can result in adverse effects, with both commonly observed in clinical settings [3]. Thus, the proper administration of opioids is crucial to meet the needs of individual patients. However, the factors contributing to different inter-individual responses to opioids are not fully understood. Moreover, morphine has been used as a potent analgesic for the treatment of severe chronic pain. However, frequent long-term treatment results in the development of analgesic tolerance [4,5,6].

Morphine produces analgesia via its activity at several levels of the nervous system; it inhibits neurotransmitter release from primary afferent terminals in the spinal cord and activates descending inhibitory controls in the midbrain [7,8,9]. However, the peripheral action of morphine in regulating pain transmission remains unclear [10]. The endogenous opioid system is activated under pathological conditions. The ratio of morphine-induced partial antinociceptive activity at peripheral sites is more advantageous than that at the central site [11,12]. Moreover, many researchers have studied the analgesic activity of several opioid receptors for a long period [13,14,15,16,17,18]. Despite extensive investigation, the detailed mechanism underlying the analgesic action of morphine is not fully understood.

In a previous study, we showed a close correlation between time-course changes in morphine-induced itching and antinociceptive activity [19]. Therefore, we suggest that the itching has a strong association with antinociceptive activity and focus on the measurement of the itch-associated scratching behavior. Itching elicits a strong desire to scratch; therefore, the scratching behavior count is a useful index to evaluate itching [20]. Using NC/Nga mice, an animal model for atopic dermatitis [21], we designed a method to assess spontaneous scratching behavior [22]. Current itch studies rely on human perception, and these nociceptive stimuli have no discernible differences. However, the sensory perception of foreign substances and true itching can be differentiated by dividing the scratching behavior of mice into long-lasting scratching (LLS; itch-associated scratching behavior) and short-lasting scratching (SLS; hygiene behavior) [22]. The number of spontaneous scratches can be automatically detected and objectively evaluated using a computer. The study showed frequent LLS in NC/Nga mice with skin lesions and not in other mouse strains. SLS was frequently observed in skin-lesioned NC/Nga mice and other strains without skin lesions. These results indicate that SLS is a form of locomotor activity and/or hygienic behavior, whereas LLS represents a true itching response in mice. The study suggests differentiating genuine itching caused by foreign substance from contact itching. This can be achieved by objective mouse scratching as LLS and SLS behaviors [22].

Interleukin-31 (IL-31) is a possible mediator of itching that induces severe pruritus and dermatitis in mice [23]. In a previous NC/Nga mouse study, we found that IL-31mRNA is expressed only in scratching mice and not in those without scratching behavior [24]. We also observed that IL-31 pretreatment induces alloknesis (itching upon touching or brushing) and hyperemesis (itchiness upon pin pricking) itching sensations around an itching site in mice [25,26,27,28]. Furthermore, morphine caused analgesia and alloknesis simultaneously [29]. Thus, in a recent study, we investigated the effect of morphine on nociception and the LLS in interleukin-31RA-deficient (IL-31RA knock-in, IL-31RAKI) and wild-type mice [19]. Morphine elicits both LLS and antinociception in wild-type mice. Additionally, a significant correlation was observed between LLS counts and antinociception, indicating a correlation between morphine-induced antinociception and itching [19]. Morphine-induced antinociception is partially reduced in IL-31RAKI mice, providing the first evidence that IL-31RA may play an important role in morphine-induced antinociception [30,31,32,33,34]. To confirm this hypothesis, we examined the effect of IL-31 pretreatment on the antinociceptive effect of morphine in mice. Furthermore, we highlighted that this effect of IL-31 resulted from an increase in the DRG neuronal level of IL-31RA via the repeated administration of IL-31, thus elucidating the site of the peripheral antinociceptive effect of morphine. Recent findings have indicated that IL-31 is partially involved in peripheral analgesic mechanisms. As IL-31 and IL-31RA are not expressed in the central nervous system [35], the endogenous opioid system is activated under pathological conditions.

Previously, we demonstrated a close correlation between morphine-induced antinociceptive activity and scratching behaviors (LLS and SLS) in mice [1]. Antinociceptive activity can only be measured intermittently; however, scratching behavior is continually measurable. Therefore, in this study, we used an index of scratching behavior to elucidate the site of action of morphine. The purpose of the present study was to investigate the effect of morphine on itch-associated scratching behavior (LLS) and antinociceptive effects in IL-31RAKI mice compared with those in wild-type mice to elucidate the regulatory mechanism of morphine-induced antinociceptive activity and antinociceptive tolerance.

## 2. Results

### 2.1. Effect of Morphine on Antinociception in Wild-Type and IL-31RAKI Mice

In the hot-plate test, no changes or significant differences were observed between the vehicle (saline, 10 mL/kg, subcutaneous) administered groups of wild-type mice (Figure 1a, blue line) and IL-31RAKI mice (Figure 1a, green line). Morphine (3 mg/kg, subcutaneous) significantly increased the latency from 15 min with peak activity at 30–60 min after administration and subsequently decreased latency to basal levels approximately 120 min after administration in wild-type and IL-31RAKI mice (Figure 1a, red and yellow lines). In wild-type mice, the antinociceptive index from 15 to 120 min after morphine administration (AUC_15–120 min_) was significantly different between saline and morphine (Figure 1b, left side). The increased latency caused by morphine administration in IL-31RAKI mice (Figure 1a, yellow line) was lower than that in wild-type mice (Figure 1a, red line). The total antinociceptive index from 15 to 120 min after morphine administration (AUC_15–120 min_) in IL-31RAKI mice was significantly lower in wild-type mice (Figure 1b, red and yellow column).

### 2.2. Effect of Single or Repeated Pretreatment with IL-31 on Morphine-Induced Antinociception in Mice

The vehicle (saline 10 mL/kg, subcutaneous) did not change antinociception during the experimental period (Figure 2a, blue line). Morphine administration (1 mg/kg, subcutaneous) significantly increased the antinociceptive activity (Figure 2a, red line) and the total antinociceptive index from 30 to 120 min (AUC 30–120 min) compared with the saline-treated group (Figure 2b, red column). A single pretreatment with IL-31 (50 μg/kg, intraperitoneal) marginally enhanced morphine-induced antinociception (Figure 2a, green line); however, the difference from the vehicle (phosphate-buffered saline, PBS)-treated group was not significant (Figure 2b, green column). In contrast, the repeated pretreatment with IL-31 (50 μg/kg, intraperitoneal, every 12 h for 3 days) significantly increased morphine-induced antinociception (Figure 2a, yellow line) and the total antinociceptive index compared to the PBS-treated group (Figure 2b, yellow column). These results suggested that the repeated administration of IL-31 is a suitable condition for evaluating the antinociceptive effects of a combination of morphine and IL-31. Therefore, we performed the following experiments with repeated intraperitoneal administration of IL-31 (50 μg/kg) every 12 h for 3 days [28].

### 2.3. Effect of IL-31 on Morphine-Induced Antinociception in Mice

The antinociceptive activity or total antinociceptive index from 30 to 90 min (AUC30–90 min) did not change in the vehicle (phosphate-buffered saline, PBS, 10 mL/kg, intraperitoneal + saline, 10 mL/kg, subcutaneous) (Figure 3a, blue line) and IL-31 (50 μg/kg, intraperitoneally, every 12 h for 3 days) + saline-treated groups (Figure 3b, blue line) compared with the saline treatment value (Figure 3c, blue and red columns). Morphine (1, 3, and 10 mg/kg, subcutaneous) delayed the latency and increased the antinociceptive activity in a dose-dependent manner (Figure 3a, red, green, yellow lines) compared to the saline-treated group. Repeated pretreatment with IL-31 enhanced the morphine-induced delay in latency and increased the antinociceptive index in a dose-dependent manner (Figure 3b, red, green, yellow lines). The total antinociceptive index from 30 to 90 min after morphine administration (AUC30–90 min) was significantly increased in mice treated with the IL-31 and morphine combination compared to those treated with the corresponding dose of the PBS and morphine combination (Figure 3c, red column).

### 2.4. Effect of Repeated Pretreatment with Three Different Doses of IL-31 on Morphine-Induced Antinociception in Mice

Only the IL-31-treated group showed no change in antinociceptive activity or total antinociceptive index (Figure 3a,b, blue line and Figure 3c saline column). Morphine administration (3 mg/kg, subcutaneous) significantly increased antinociception (Figure 4a, red line) compared with the saline-administered group (Figure 4a, blue line). Additionally, the total antinociceptive index from 30 to 120 min after morphine administration (AUC_30–120 min_) in the morphine + saline group was significantly higher than that in the saline + vehicle (phosphate-buffered saline, PBS)-treated group (Figure 4b, red column). Repeated pretreatment with IL-31 (5, 15, and 50 μg/kg, intraperitoneal, every 12 h for 3 days) increased morphine-induced antinociception in a dose-dependent manner (Figure 4a, green, yellow, purple lines). Moreover, the total antinociceptive index from 30 to 120 min after morphine administration (AUC_30–120 min_) of the IL-31-treated groups was significantly higher than that of the morphine + PBS-treated group (Figure 4b, yellow and purple columns).

### 2.5. Effect of Morphine on Scratching Behavior in Wild-Type and IL-31RAKI Mice

Morphine administration (3 mg/kg, subcutaneous) increased scratching behavior, both LLS and SLS, from 15 to 120 min with peak activity at 30–60 min, and subsequently decreased to basal levels approximately 120 min after morphine administration in wild-type mice (Figure 5a,b). Morphine-induced total LLS counts for 2 h were significantly suppressed in IL-31RAKI mice (Figure 5c, red and yellow column). However, morphine-induced SLS counts were significantly increased in wild-type and IL-31RAKI mice (Figure 5d, red and yellow columns). A close correlation has been previously observed between morphine-induced LLS counts and antinociception [12]. These data indicate that morphine-induced LLS is a result of peripheral nervous action and morphine-induced SLS is a reaction derived from the central nervous system. Two experiments were performed to elucidate the regulatory mechanism of morphine-induced LLS counts as an indicator of peripheral antinociceptive activity.

### 2.6. Effect of Morphine-Induced LLS Counts in Naïve or Mite-Infested Mice

In morphine administered naïve mice (3 mg/kg, subcutaneous), the LLS counts gradually increased from 30 min, reached a peak at 60 min, and then decreased to basal levels after 180 min (Figure 6a, blue line). This increase in LLS count was similar in both naïve and mite-infested (MI, cohoused with skin-lesioned NC/Nga mice) mice (Figure 6a, red line). The total LLS counts determined at 2 h were not significantly different (Figure 6b). The total LLS counts determined for 24 h before morphine administration were significantly higher in mice with a mite infestation (MI) than those in naive mice (Figure 6c). The expression levels of IL-31RA in the skin (Figure 6d) and DRG (Figure 6e) were also significantly higher in mice with a mite infestation (MI) than those in naïve mice. The serum IL-31 concentration was not detected in either mite-infested or naïve mice (Figure 6f).

### 2.7. Effect of Intravenous IL-31 Injection on Morphine-Induced LLS Counts in Mice

In vehicle (phosphate-buffered saline, PBS, 10 mL/kg, intravenous)-injected mice, morphine administration (3 mg/kg, subcutaneous) gradually increased the LLS counts (Figure 7a, blue line). This increase in the LLS count was significantly enhanced by pretreatment with a single intravenous injection of IL-31 (1 mg/kg, Figure 7a, red line). Specifically, the total LLS counts significantly increased for 2 h after morphine administration following pretreatment with IL-31 (Figure 7b). An administration of IL-31 (1 mg/kg, intravenous) before morphine administration did not change the LLS counts (Figure 7c), cutaneous IL-31RA expression (Figure 7d), or DRG neuronal IL-31RA expression (Figure 7e). However, the serum IL-31 concentration was markedly different. Statistical processing was not possible, and the differences were not significant (Figure 7f).

### 2.8. Effect of a Single Dose of Morphine on Spontaneous Scratching Behavior in Mice

Spontaneous scratching behavior had a circadian rhythm, and LLS and SLS increased during the dark phase (19:00–7:00, shaded area), especially from 19:00 to 24:00 (Figure 8a,b). The spontaneous LLS (Figure 8a, red line) and SLS (Figure 8c, red line) counts decreased 6 h after morphine pretreatment (5 mg/kg, subcutaneous). The total counts of LLS (Figure 8c, red column) and SLS (Figure 8d, red column) for 12 h from 6 h after morphine administration significantly decreased upon morphine pretreatment compared to those of the saline pretreatment group. Single morphine administration resulted in a decrease in scratching counts (LLS and SLS) over an extended period. Next, we examined the influence of repeated morphine administration on antinociceptive activity, which is closely related to the scratching behavior.

### 2.9. Effects of Repeated Morphine Administration of on Morphine-Induced Scratching Behavior and Antinociceptive Activity in Mice

There were no changes in the LLS and SLS in the saline repeated administration (every 3 h, 10 mL/kg, subcutaneous) group, except in the dark phase of the circadian rhythm (19:00–24:00, Figure 9a, blue line). In contrast, the first morphine (5 mg/kg, every 3 h, subcutaneous) administration increased the LLS, and the SLS peaked at 1 h and decreased 3 h after administration (Figure 9a, red line). The repeated administration of morphine every 3 h gradually decreased the morphine-induced LLS and SLS. A significant decrease was observed in morphine-induced total LLS (Figure 9b) and SLS (Figure 9c) counts in the third (Figure 9b,c, red columns) and ninth (Figure 9b,c, green columns) morphine administration groups compared with the first morphine administered group (Figure 9b,c, blue columns). The antinociceptive effect of repeated administration of morphine also decreased in the third (Figure 9d, red column) and ninth (Figure 9d, green columns) morphine-administered groups compared with that of the first morphine-administered value (Figure 9d, blue column). Given that the LLS counts gradually decreased after consecutive repeated administrations of morphine, we examined the effect of an intravenous injection of IL-31 3 h after the last dose of morphine and measured the reactivity of the IL-31-induced LLS counts.

The effect of IL-31 on LLS counts after the ninth administration of morphine or saline were investigated. IL-31 (1 mg/kg, intravenous injection) was administered 3 h after the last dose of these administration In the repeated saline-administered group, the LLS counts were markedly increased (Figure 9a, blue line). However, repeated administrations of morphine completely suppress the LLS counts (Figure 9a, red line, after blue arrow). The total LLS counts for the IL-31 administration (after 0–12 h) were significantly lower in the morphine-administered group than that in the saline-treated group (Figure 9e).

### 2.10. Effect of Combination Treatment of IL-31 and Morphine on Morphine-Induced Antinociceptive Tolerance and DRG Neuronal IL-31RA Expression in Mice

There was no change in the antinociceptive activity in the vehicle (phosphate-buffered saline, PBS, 10 mL/kg) or IL-31 (50 μg/kg, intraperitoneal) on vehicle (saline, 10 mL/kg, subcutaneous) administration once a day (9:00) for 5 days. In contrast, morphine (10 mg/kg, subcutaneous, once a day for 5 days) gradually decreased the morphine dosage required for morphine-induced antinociceptive activity (Figure 10a left side). There was a significant difference compared with the values of first-morphine administration-induced antinociceptive activity at 3–5 days after morphine administration (Figure 10a, left side, green, yellow, and purple columns). The decrease in antinociceptive activity is believed to be the result of antinociceptive tolerance induced by the repeated administration of morphine. In contrast, the combination treatment of IL-31 (50 μg/kg, intraperitoneally, every 12 h for 5 days) also gradually decreased the degree of morphine dosage on morphine-induced antinociceptive activity (Figure 10a, right side). No significant difference was observed before morphine administration-induced antinociceptive activity during the experimental period (Figure 10a, right side).

There was no change in the repeated morphine administration (10 mg/kg, subcutaneous, once a day for 5 days) on DRG neuronal IL-31RA expression levels (Figure 10b, blue line). In contrast, combined administration of morphine and IL-31 (50 μg/kg, intraperitoneally, every 12 h for 5 days) gradually increased the DRG neuronal IL-31RA expression (Figure 10b, red line).

## 3. Discussion

Previously, we reported that antinociceptive activity of several analgesics significantly increase by pretreatment of IL-31 [29]. Moreover, when combined with morphine, the analgesic effect was enhanced. We suggested that IL-31 played a role in the antinociceptive action of morphine [19]. We examined the effect of IL-31 pretreatment on morphine-induced antinociception using a hot-plate test. There was a close correlation between morphine-induced LLS counts and antinociceptive effects. We investigated LLS as an indicator of morphine-induced antinociceptive activity. These data showed that cutaneous-injected IL-31-induced LLS was enhanced by DRG neuronal IL-31RA expression [36]. Morphine treatment significantly increased the IL-31-induced LLS and antinociceptive activity. The subcutaneous injection of morphine induced LLS (itch-associated scratching behavior), SLS (hygiene behavior), and antinociception. The mechanism of morphine antinociception involves two sites of action: central and peripheral [37,38]. Previously, we have reported that IL-31 may play a more significant role in the modulation of peripheral morphine-induced antinociception via sensory neurons in IL-31RAKI mice than in wild-type mice [19]. Antinociception and scratching behaviors (LLS and SLS) were observed simultaneously during the experimental period. In particular, a close correlation was observed between the development of scratching behavior (LLS and SLS) and morphine-induced antinociceptive action. These results showed that these behaviors mediate increased itching, locomotion, and antinociceptive activity. As LLS expression is not frequently observed in other pruritogen-induced scratching behaviors [39,40], IL-31RA may be involved in IL-31-induced LLS expression [28]. Therefore, we generated IL-31RAKI mice and examined the effect of morphine on LLS (itching) and its antinociceptive activity in comparison with wild-type mice. We observed that morphine-induced LLS did not develop in IL-31RAKI mice and that morphine-induced antinociceptive effects were partially observed. Collectively, these results indicated that morphine-induced SLS and partial antinociception are central site actions, whereas morphine-induced LLS (itching) and partial antinociception are peripheral site actions. The peripheral site activity of morphine in the regulation of pain transmission is unclear; however, the endogenous opioid system is activated under pathological conditions. Therefore, the mechanism underlying the analgesic effects causing itching remains unclear.

Innocuous mechanical stimuli that do not normally induce a behavioral response in mice elicit bouts of scratching following intradermal administration of the pruritogen histamine, 5-HT, and PAR-4 agonist. Similarly, following the local application of histamine or 5-HT in humans, innocuous mechanical stimuli induced itching, a phenomenon known as alloknesis [41,42,43,44]. In contrast, post-histamine mechanically induced scratching in mice was significantly attenuated by the μ-opioid antagonist naltrexone, which is consistent with the ability of another antagonist, naloxone, to reduce alloknesis in humans [45]. Moreover, morphine potentiates histamine-induced itching in humans and chloroquine-induced scratching in rats [46]. These results suggest that morphine (opioid) administration causes hyperknesis and alloknesis.

Recently, we reported that pretreatment with IL-31 causes alloknesis (LLS) and antinociceptive activity simultaneously in mice [19]. In contrast, morphine-induced LLS and antinociceptive effects were closely correlated in wild-type mice. Repeated pre-administration of IL-31 marginally increased the morphine-induced antinociceptive effects in the hot-plate test. Moreover, repeated administration of IL-31 significantly increased LLS counts and the IL-31RA mRNA expression in the cervix and sacral plexus of DRG neuron cell bodies [36]. These data suggest that IL-31 upregulates the IL-31RA expression in DRG neuron cell bodies, enhancing topical cutaneous IL-31-induced LLS via IL-31RA levels [30]. These results suggest that the individuality of analgesic sensitivity to morphine was due to differences in the interneural IL-31RA expression. Thus, LLS and the antinociceptive activity depend on the IL-31RA expression in the DRG and the concentration of IL-31 in the blood [30]. Our hypothesis links IL-31 and morphine via a potential site that promotes IL-31RA exocytosis, thus reinforcing the antinociceptive activity of morphine. Rapidly increasing IL-31 and IL-31RA synthetically is impossible, and the release of trace IL-31 into the blood remains consistent. Hence, we studied two morphine types under neural conditions to discern if the site of action was the pre-synaptic or post-synaptic sensory nervous system.

Mite-infestation-induced itching in mice was caused by co-housing mice with skin-lesioned NC/Nga mice. Several strains of mice showed increased LLS counts after only a few days of co-housing with skin-lesioned NC/Nga mice [47]. Mite infestation significantly increased DRG neuronal IL-31RA expression and not the IL-31 blood concentration (Figure 11a, upside). Based on these data, we examined the site of action of morphine, considering the enhancement of morphine-induced LLS counts as an indicator of antinociceptive activity following pretreatment with IL-31. Mite-infested mice demonstrated a significantly increased LLS count; however, this effect was difficult to observe during the day as this reaction occurred at night [28]. Although morphine-induced LLS counts did not change in mite-infested mice, DRG neuronal IL-31RA increased (Figure 11a, downside). These results suggest that morphine did not affect the release or production of endogenous IL-31.

In BALB/c mice, a pretreatment with IL-31 1 h before morphine administration did not increase the level of IL-31RA expression in the skin or neuronal DRG; however, it markedly increased the blood concentration of IL-31 (Figure 11b, upside). Morphine-induced LLS counts, as an indicator of antinociceptive activities, were enhanced compared with those in the PBS-treated group (Figure 11b downside). The site of action of morphine is the pre-synaptic sensory nervous system. The antinociceptive activity of morphine was partially inhibited in IL-31RAKI mice. The IL-31RA effect disappears on frequent administration of morphine. Repeated morphine administration had no effect on DRG neuronal IL-31RA expression levels. From these results, we suggest that morphine may accelerate the ability of IL-31RA to act on sensory nerves (e.g., promotion of the translocation of IL-31RA to the plasma membrane and exocytosis of IL-31RA). For example, intra neural catecholamines appear to be increased after reserpine administration for approximately 2 h, increasing the sympathetic nerve activity.

Genetic factors may also affect individual differences in opioid sensitivity [1]. Previous reports have shown a reduced sensitivity to opioids in heterozygous μ-opioid receptor knockout mice with 50% μ-opioid receptor mRNA [48,49]. This suggested that the low amounts of μ-opioid receptor mRNA cause a reduced sensitivity to opioids. Moreover, interindividual differences in opioid analgesia may be partly attributable to divergent μ-opioid receptor mRNA levels because of μ-opioid receptor gene differences [50]. The mechanism by which they show inter-individual differences in morphine is regarded as a phenomenon occurring in the central nervous system. In contrast, this study showed that the inter-individual differences in morphine levels in the peripheral nerve depend on the difference in the levels of intraneural IL-31RA. Additionally, neuronal IL-31RA expression in the DRG may contribute to individual differences in morphine analgesic sensitivity. Repeated administration of IL-31 gradually increased the LLS counts, DRG neuronal IL-31RA mRNA expression, and morphine-induced antinociception. These findings suggest that neuronal IL-31RA expressed in the DRG is involved in the modulation of morphine-induced antinociception. Furthermore, morphine may induce exocytosis of IL-31RA at the end of the sensory nerve. Thus, IL-31 may be a useful adjunctive agent for pain relief in combination with morphine. Therefore, the combination treatment with IL-31 and morphine may reduce the required dose of morphine and suppress the onset of side effects. Furthermore, this combination may reduce individual differences in morphine sensitivity.

In this study, a single dose of morphine significantly decreased spontaneous scratching behavior over 6 h (dark phase, 19:00–24:00) after morphine administration. Moreover, nine-times-repeated administration of morphine also decreased the morphine-induced scratching behavior and antinociceptive activity, with every degree of morphine administration. Moreover, IL-31-induced LLS counts were significantly reduced after nine-times-repeated administration of morphine (every 3 h for 27 h). This phenomenon was regarded as the result of intraneural IL-31RA depletion caused by the repeated administration of morphine-induced exocytosis of IL-31RA. Previously, we have reported that IL-31RA may play a crucial role in sensory neurons in peripheral morphine-induced antinociception modulation in IL-31A deficient mice. In addition, they closely correlated with peripheral activity, morphine-induced SLS counts, and central antinociceptive activity. Therefore, we suggest that morphine-induced antinociceptive tolerance was caused by both peripheral and central sites of action, as morphine decreased the LLS and SLS counts after a single or repeated administration. Moreover, the recovery time of morphine-induced tolerance formation, the central site of action was more than 24 h, and the peripheral site of action was more than 12 h after the last dose of nine-times-repeated administration of morphine.

To date, several studies have investigated the mechanisms underlying morphine-induced antinociceptive tolerance. NMDA receptors [51], cannabinoid receptors [52], transient receptor potential vanilloid-1 [53], the serotonergic system, and the noradrenergic system [54,55,56] play a substantial role in morphine-induced tolerance. However, the exact mechanism underlying morphine-induced tolerance remains unknown. Previously, we have reported that pretreatment with IL-31 significantly enhances morphine-induced antinociceptive activity, and that the site of action of morphine may promote exocytosis of IL-31RA in peripheral sensory nerve endings (Figure 12a, upside). Therefore, repeated administration of morphine depletes IL-31RA in the peripheral sensory nerves through the recurrent promotion of IL-31RA exocytosis (Figure 12a, middle side). Repeated morphine administration decreases morphine-induced LLS and the antinociceptive activity (Figure 12a, downside). For example, intraneural catecholamine depletion after reserpine administration for approximately 24 h decreases the sympathetic nerve activity.

Repeated doses of IL-31 administered every 12 h for 3 days significantly increased the LLS counts and IL-31RA expression in the DRG neuron cell body, regardless of the IL-31 dosage [30]. In contrast, repeated administration of morphine did not influence DRG neuronal IL-31RA expression (Figure 12b, upside). These data suggest that IL-31 upregulates IL-31RA expression in DRG neurons, and that topical cutaneous IL-31-induced LLS (itching) is enhanced by IL-31RA in the same topically innervated DRG neuron (Figure 12b, middle side). These data show that itching depends on the interaction between DRG neuronal IL-31RA and blood IL-31 concentrations [35]. Therefore, combining the IL-31 and morphine treatments could potentially address IL-31RA depletion in sensory nerves (Figure 12b, downside). Thus, we examined the effect of a combined treatment with morphine and IL-31 on repeated administration of morphine to induce antinociceptive tolerance in mice, which resulted in decreased antinociceptive activity. The increase significantly decreased with increasing morphine doses. In contrast, repeated combined administration of IL-31 and morphine gradually decreased the antinociceptive activity; however, the difference was not significant (Figure 12b, downside). The antinociceptive tolerance of morphine involves dual actions; one shifts pain to itch via IL-31-induced alloknesis at the peripheral site, whereas the other triggers inhibitory controls at the central site. It has been suggested that IL-31 acts only at peripheral sites of morphine-induced analgesic tolerance. Moreover, clinical evidence has revealed that opioid tolerance does not develop frequently and that opioid treatments are not always problematic in terms of creating tolerance in patients with chronic pain [57,58,59]. Recently, we reported that DRG neuronal IL-31RA expression increases in pain-induced model mice [27]. The results explained that analgesic-tolerance relaxes in patients with chronic pain. These results suggest that morphine is united with IL-31RA, which contributes to antinociceptive activity rather than a former point of action for morphine and the relations of IL-31. These results suggest that the repeated administration of IL-31 partially modulates nociception and tolerance to morphine antinociception. However, the function of opioid receptors and IL-31RA and the relationship between their existence are unknown. Therefore, further studies are warranted.

In conclusion, we suggest that IL-31RA may play a critical role in morphine-induced peripheral antinociception. In contrast, a combination treatment with IL-31 and morphine significantly increased the morphine-induced antinociceptive activity. IL-31 upregulates IL-31RA expression in DRG neurons and morphine-induced antinociceptive activity is dependent on DRG neuronal IL-31RA expression. These results suggest that the individuality of analgesic sensitivity to morphine was caused by differences in the expression of intraneural IL-31RA. Therefore, a pretreatment of IL-31 may reduce individual differences in the antinociceptive activity of morphine. The combination treatment with IL-31 and morphine may reduce the required dose of morphine and suppress the onset of its side effects. This site of action of morphine may induce the exocytosis of IL-31RA in the sensory nervous system. Repeated administration of morphine decreased the morphine-induced scratching behavior (LLS and SLS) and antinociception (analgesic tolerance). Repeated administration of morphine completely inhibited IL-31-induced LLS (itching). These results suggest that the repeated administration of morphine depletes inter-neuronal IL-31RA levels. Repeated administration of IL-31 increased IL-31RA expression in DRG. Furthermore, its combination with morphine enhanced its analgesic action. Collectively, the suppressive effect of IL-31 on morphine-induced antinociceptive tolerance may result, at least in part, from IL-31RA supplementation in the sensory nerves. These results suggest that the combination of morphine and IL-31 reduces the analgesic tolerance of morphine.

## 4. Materials and Methods

### 4.1. Animals

Male skin-lesioned-NC/Nga mice aged 10–13 weeks and BALB/c and C57BL/6 mice aged 6–8 weeks were purchased from SLC Japan (Shizuoka, Japan). We used the NC/Nga mice to increase IL-31RA levels in BALB/c mice. Mice lacking IL-31RA (IL-31RA^−/−^) were generated as described previously. The IL-31RAKI (IL-31RA-deficient) mice used in this study were obtained from a C57BL/6 genetic background and hybrid mutant mice were originally created based on a 129 SVJ-C57BL/6 background by backcrossing breeding over 15 generations [35]. In this study, we used age-matched male homozygous (IL-31RAKI, IL-31RA^−/−^) and wild-type (IL-31RA^+/+^) mice. C57BL/6 mice were used as the target group in the experiments using IL-31RAKI mice. The animals were housed under controlled temperature (23 ± 3 °C), humidity (50 ± 5%), and lighting (lights on from 7:00 a.m. to 7:00 p.m.). All animals had free access to food and tap water. All animal experiments were approved by the Committee for Animal Experimentation at the International University of Health and Welfare, in accordance with the Guidelines for Proper Conduct of Animal Experiments (Science Council of Japan, 2006).

### 4.2. Drugs

Mouse IL-31 cDNA-spanning amino acids 24–163 of IL-31 were cloned in a frame with pET30A (Novagen, Darmstadt, Germany), and the construct was transformed into BL-21 cells (Novagen). After induction with isopropyl-β-D-thiogalactopyranoside, IL-31 was purified under denaturing conditions using nickel-chelating Sepharose (Qiagen, Benelux B.V., Hulsterweg, Netherlands) and dialyzed in phosphate-buffered saline (PBS) [35]. IL-31 at a dose of 50 μg/kg was used for subcutaneous or intravenous injections, as previously described [35]. Morphine hydrochloride (Takeda Pharmaceutical Co., Ltd., Osaka, Japan) was dissolved in saline and administered subcutaneously at 0.5 mL/kg.

### 4.3. Hot-Plate Test

The hot-plate test [60] was used to measure withdrawal latency, as described previously [16]. Mice were placed on a hot-plate maintained at 50 ± 0.5 °C, and the latency to either a paw-lick or an attempt to escape by jumping was recorded. To prevent tissue damage, the mice that showed no response within 60 s were removed from the hot-plate and assigned a score of 60 s. The percentage of nociception (nociceptive index) was calculated using the following formula: [(*T*_1_ − *T*_0_)/(*T*_2_ − *T*_0_)] × 100, where *T*_0_ and *T*_1_ are the latencies observed before and after drug administration, respectively, and *T*_2_ is the cut off time (60 s). The animals were tested before and 30, 60, 90, 120, 150, and 180 min after drug administration.

### 4.4. Measurement of Scratching Counts

Scratching counts were measured as previously described [19]. A small magnet (diameter, 1.0 mm; length, 3.0 mm) was implanted subcutaneously into both hind paws of isoflurane-anesthetized mice 24 h before the measurement. Each mouse was placed in an observation chamber (diameter, 11 cm; height, 18 cm) surrounded by a circular coil by which an electric current, induced by the movement of the magnets attached to the hind paws, was amplified and recorded. The number of spontaneous scratches was automatically detected and objectively evaluated using a computer with MicroAct (Neuroscience, Tokyo, Japan) [61]. The analysis parameters for detecting waves were as follows: threshold, 0.1 V; event gap, 0.2 s; minimum duration, 0.3 or 1.0 s; maximum frequency, 20 Hz; and minimum frequency, 2 Hz.

### 4.5. Measurement of Serum IL-31 Concentration

Approximately 0.2 mL of blood was collected from the femoral vein of mice under isoflurane anesthesia. Serum samples were obtained via centrifugation (3000 rpm, 10 min) and stored at −5 °C until assayed. The IL-31 concentration in the blood was measured using a sandwich ELISA. Briefly, 96-well ELISA plates were coated with 100 µL diluted capture anti-body (1 µg/mL anti-IL-31 rabbit polyclonal antibody) over night at 4 °C. The wells were aspirated, washed thrice, and incubated for 2 h at 25 ± 5 °C with blocking buffer (200 µL 0.5% bovine serum albumin). After aspiration, wells were washed thrice and incubated for 2 h at 25 ± 5 °C with 100 µL of standard or test sample. The wells were aspirated once more, washed five times, and incubated for 1 h at 25 ± 5 °C with 100 µL detection anti-body (1 µg/mL biotinylated anti-IL-31RA hamster monoclonal antibody). After aspiration, wells were washed five times and incubated for 30 min at 25 ± 5 °C with 100 µL streptavidin-horseradish peroxidase conjugates. Following aspiration, wells were washed seven times and incubated for 30 min at 25 ± 5 °C in the dark with 100 µL tetramethylbenzidine substrate solution. Following incubation, 50 µL H_2_SO_4_ 2N stop solution was added to each well, and the absorbance was read at 450 nm within 30 min with λ correction at 570 nm. All samples were tested in duplicate. The sensitivity and specificity of the IL-31 assay were 1.0 ng/mL.

### 4.6. Real-Time Quantitative PCR (RT-PCR)

The gene expression levels of IL-31, IL-31RA, and β-actin were measured using real-time polymerase chain reaction (RT-PCR) in DRG (C_4–7_, T_1–4_) [62] neuronal cell bodies from the shoulders and backs of BALB/c and C57BL/6 mice at each point. Total RNA was extracted from the dorsal skin of each mouse using the Triazole reagent (Invitrogen, Carlsbad, CA, USA) and digested using amplification-grade DNase I (Invitrogen) according to the manufacturer’s instructions. cDNA was synthesized by the SuperScript III First-Strand Synthesis System (Invitrogen). Quantitative RT-PCR was performed using SYBR Green Master Mix and the Applied Biosystems 7700 Sequence Detection System (Applied Biosystems, Foster City, CA, USA). PCR primers for IL-31 were designed using PRIMER 3 software (V. 0.4.0), and primers for β-actin were purchased from TAKARA BIO (Otsu, Shiga, Japan). Primer sequences were as follows: IL-31 (5′-ATA CAG CTG CCG TGT TTC AG-3′ and 5′ -AGC CAT CTT ATC ACC CAA GAA-3′), IL-31RA (5′-CCA GAA GCT GCC ATG TCG AA-3′ and 5′-TCT CCA ACT CGG TGT CCC AAC-3′), and β-actin (5′ -TGA CAG GAT GCA GAA GGA GA-3′ and 5′-GCT GGA AGG TGG ACA GTG AG-3′). The relative expression levels were calculated using the relative standard curve method outlined in the manufacturer’s technical bulletin. A standard curve was generated using fluorescence data obtained from four-fold serial dilutions of the total RNA of the sample with the highest expression. The curve was used to calculate the relative amounts of the target mRNA in the test samples. The quantities of all targets in the test samples were normalized to the corresponding β-actin RNA transcripts in the skin samples.

### 4.7. Statistical Analyses

All data were analyzed using GraphPad InStat and GraphPad Prism (GraphPad Software, Version 7, Inc., La Lolla, CA, USA). The experimental values are expressed as the mean and standard errors. Data on time-course changes in scratching counts, latency, or the percentage of antinociception were analyzed using two-way analysis of variance (ANOVA) followed by the Bonferroni test. One-way ANOVA followed by the Student–Newman–Keuls multiple comparison test, was used to compare other data. To investigate the correlation between the time-course change in the antinociceptive effect and scratching behavior induced by morphine, the latency or percentage of antinociception and LLS or SLS at each time point after administration of morphine were plotted, and the product moment correlation coefficient was calculated. Values of *p* < 0.05 were considered statistically significant.

## 5. Conclusions

We suggest that IL-31RA plays an important role in morphine-induced peripheral antinociception. Pretreatment with IL-31 enhanced the morphine-induced antinociceptive activity. The mechanism of action of morphine may accelerate the ability of IL-31RA to easily act on the sensory nerves. This site of action of morphine may induce exocytosis of IL-31RA in the sensory nervous system. Therefore, pretreatment with IL-31 may reduce the use of morphine, several adverse reactions, and individual differences in its antinociceptive activity. To further explore the beneficial effects of IL-31, we examined its influence on morphine-induced antinociceptive tolerance in mice. Repeated administration of IL-31 increased IL-31RA expression in the dorsal root ganglia. Furthermore, its combination with morphine enhanced its analgesic action. We aimed to elucidate the mechanism by which morphine-induced antinociception is modulated by DRG neuronal IL-31RA expression. Collectively, the suppressive effect of IL-31 on morphine-induced antinociceptive tolerance may result, at least in part, from IL-31RA supplementation in the sensory nerves. The combination of morphine and IL-31 may reduce adverse reactions and analgesic tolerance to morphine.

## Figures and Tables

**Figure 1 ijms-24-16548-f001:**
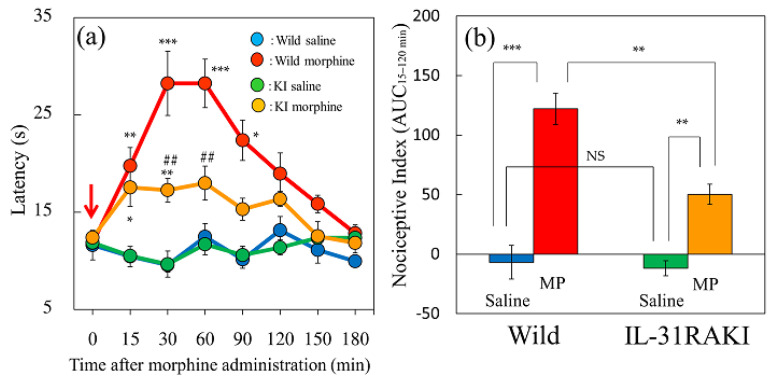
Effect of morphine on latency and antinociceptive activity in wild-type and IL-31RAKI mice. (**a**) Time-course change in latency following morphine (3 mg/kg, subcutaneous) administration in wild-type and IL-31RA KI mice. The blue line indicates the saline-administered group in wild-type mice; the green line indicates the saline-administered group in IL-31RAKI mice; the red line indicates the morphine-administered group in wild-type mice; the yellow line indicates the morphine-administered group in IL-31RAKI mice. Red arrows indicate saline or morphine administration point. Each value represents the mean ± S.E. of six mice. * *p* < 0.05, ** *p* < 0.01, and *** *p* < 0.001 compared with corresponding values of saline-administered groups. ^##^ *p* < 0.01 compared with the values of morphine-administered groups in wild-type mice. (Student’s *t*-test with Bonferroni correction). (**b**) Antinociceptive activity from 15 to 120 min after morphine administration. The blue column indicates the saline-administered group in wild-type mice; the green column indicates the saline-administered group in IL-31RAKI mice; the red column indicates the morphine-administered group in wild-type mice; the yellow column indicates the morphine-administered group in IL-31RAKI mice. IL-31, interleukin-31; Wild, C57BL/6 mice; IL-31RAKI, IL-31RA deficient mice (C57BL/6 genetic background). Each value represents the mean ± S.E. of six mice (total, 24 mice). NS, not significant. ** *p* < 0.01, and *** *p* < 0.001 compared with each value of the wild-type and IL-31RAKI mice group (Tukey’s multiple comparison test).

**Figure 2 ijms-24-16548-f002:**
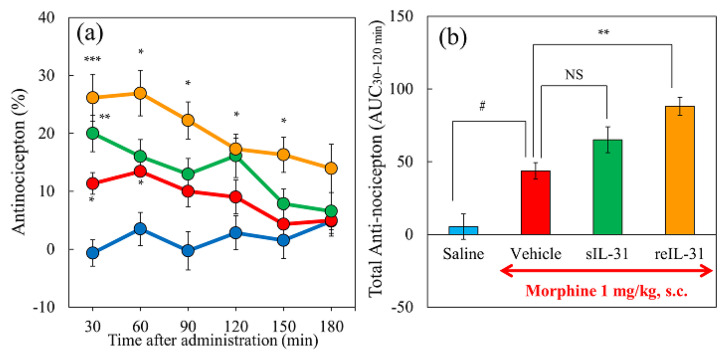
Effects of single or repeated pretreatment with IL-31 on morphine-induced antinociceptive activity in mice. (**a**) Time-course change of morphine-induced (1 mg/kg, subcutaneous) antinociceptive activity in mice using the hot-plate test. The results are presented as the mean ± S.E. of six mice. * *p* < 0.05, ** *p* < 0.01, and *** *p* < 0.001 compared with corresponding values of the vehicle (phosphate-buffered saline, PBS, 10 mL/kg, intraperitoneal) + saline-treated group (Student’s *t*-test with Bonferroni correction). (**b**) Total antinociceptive index (antinociceptive activity from 30 to 120 min (AUC30–120 min). sIL-31, single administration of IL-31 (50 mg/kg, intraperitoneal); reIL-31, repeated administration of IL-31 (50 μg/kg, intraperitoneal, every 12 h for 3 days). For both panels, the blue line and column indicate the PBS + saline-treated group; red line and column indicate the vehicle + morphine-treated group; green line and column indicate the single IL-31 + morphine treated-group; and yellow line and column indicate repeated IL-31 + morphine treated-group. The results are presented as the mean ± S.E. of six mice (total, 24 mice). ^#^ *p* < 0.05 compared with the PBS + saline-treated group (Student’s *t*-test). NS, not significant. ** *p* < 0.01 compared with the values of the PBS + morphine-treated group (Dunnett test).

**Figure 3 ijms-24-16548-f003:**
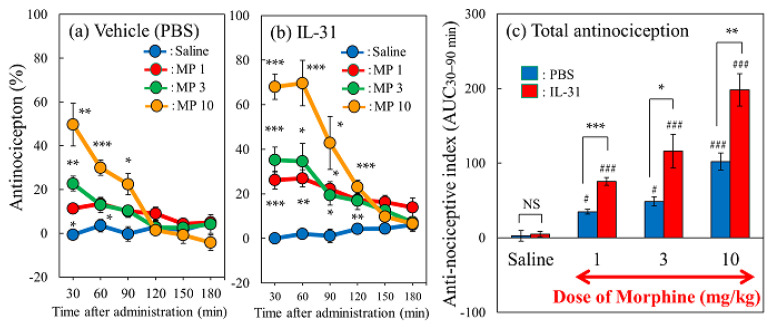
Effect of IL-31 on morphine-induced antinociceptive activity in mice. (**a**) Time-course change of morphine-induced antinociception + vehicle (phosphate-buffered saline, PBS, 10 mL/kg, intraperitoneally, every 12 h for 3 days) in hot-plate test. The blue line indicates the PBS + saline-treated group; the red line indicates the PBS + morphine 1 mg/kg-treated group; the green line indicates the PBS + morphine 3 mg/kg-treated group; the yellow line indicates the PBS + morphine 10 mg/kg-treated group. (**b**) Time-course change of three doses of morphine-induced antinociception with repeated administration of IL-31 (50 μg/kg, intraperitoneally, every 12 h for 3 days). The blue line indicates the repeated PBS + saline-treated group; the red line indicates the IL-31 + morphine (1 mg/kg, subcutaneous)-treated group; the green line indicates the repeated IL-31 + morphine (3 mg/kg, subcutaneous)-treated-group; and the yellow line indicates the repeated IL-31 + morphine (10 mg/kg, subcutaneous)-treated-group. The results are presented as the mean ± S.E. of six mice; * *p* < 0.05, ** *p* < 0.01, and *** *p* < 0.001 compared with corresponding values of the PBS + saline-treated group (Student’s *t*-test with Bonferroni correction). (**c**) Combined total antinociceptive index from 30 to 90 min (AUC_30–__9__0 min_) of morphine and IL-31. The blue column indicates the vehicle + saline- or morphine-treated groups; the red column indicates the IL-31 + morphine-treated groups. The results are presented as the mean ± S.E. of six mice (total, 48 mice). NS, not significant. * *p* < 0.05, ** *p* < 0.01, and *** *p* < 0.001 compared with the corresponding dose of PBS + morphine-treated group (Student’s *t*-test). ^#^
*p* < 0.05 and ^###^ *p* < 0.001 compared with the corresponding values of the PBS + saline-treated or IL-31 + saline-treated group (Student’s *t*-test).

**Figure 4 ijms-24-16548-f004:**
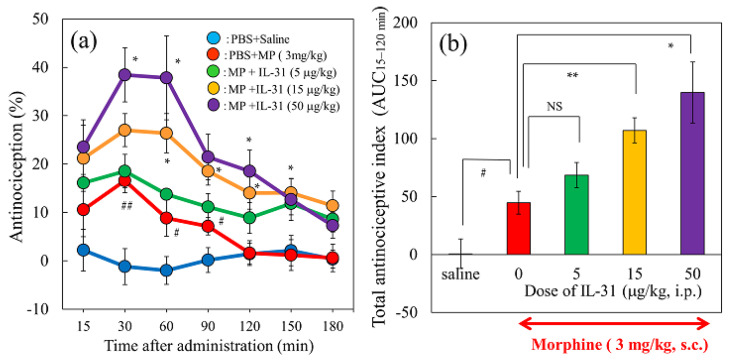
Effect of repeated pretreatment with three doses of IL-31 on morphine-induced antinociception in mice. (**a**) Time-course change of morphine (3 mg/kg, subcutaneously)-induced antinociception with three doses of repeated administration of IL-31 (5, 15, and 50 μg/kg, intraperitoneal). The results are presented as the mean ± S.E. of six mice. ^#^ *p* < 0.05, ^##^ *p* < 0.01 compared with corresponding values of the saline-treated group. * *p* < 0.05 compared with the values of the vehicle (phosphate-buffered saline, PBS, 10 mL/kg, intraperitoneal) + morphine-treated group (Student’s *t*-test with Bonferroni correction). (**b**) Effects of pretreatment with repeated administration of IL-31 on morphine-induced total antinociception index from 30 to 120 min after morphine administration (AUC_15–120 min_) in the hot-plate test. The blue line and column indicate the repeated PBS + saline (10 mL/kg, subcutaneous)-treated group; the red line and column indicate the repeated PBS + morphine-treated group; the green line and column indicate the repeated IL-31 (5 μg/kg, intraperitoneal) + morphine-treated group; the yellow line and column indicate the repeated IL-31 (15 μg/kg, intraperitoneal) + morphine-treated group; and the purple line and column indicate the repeated IL-31 (50 μg/kg, intraperitoneal) + morphine-treated group. The results are presented as the mean ± S.E. of six mice (total, 30 mice). NS, not significant. ^#^ *p* < 0.05 compared with the PBS + saline-treated group (Student’s *t*-test). * *p* < 0.05 and ** *p* < 0.01 compared with the values of the PBS + morphine-treated group (Dunnett test).

**Figure 5 ijms-24-16548-f005:**
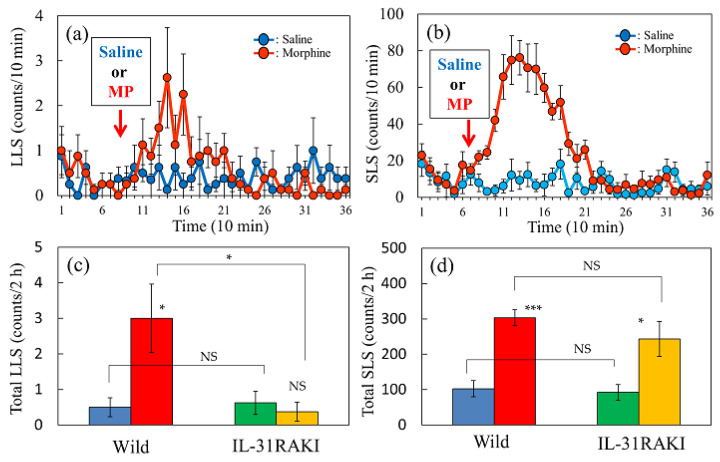
Effect of morphine on LLS and SLS counts in wild-type and IL-31RAKI mice. (**a**) Time-course changes in LLS after administration of morphine (MP, 3 mg/kg, subcutaneous) in wild-type mice. (**b**) Time-course changes in SLS after administration of morphine in wild-type mice. (**c**) Total LLS counts (counts/24 h) following saline- and morphine-administration in wild-type and IL-31RAKI mice. (**d**) Total SLS counts (counts/24 h) following saline and morphine administration in wild-type and IL-31RAKI mice. IL-31, interleukin-31; Wild, C57BL/6 mice; IL-31RAKI, IL-31RA deficient mice (C57BL/6 genetic background). The red arrows indicate the saline or morphine administration point. LLS, itch-associated scratching behavior; SLS, hygiene behavior. Values represent the mean ± S.E. of six mice (total 24 mice). NS, not significant, * *p* < 0.05, and *** *p* < 0.001 compared with each value of wild-type and IL-31KI mice (Tukey’s multiple comparison test).

**Figure 6 ijms-24-16548-f006:**
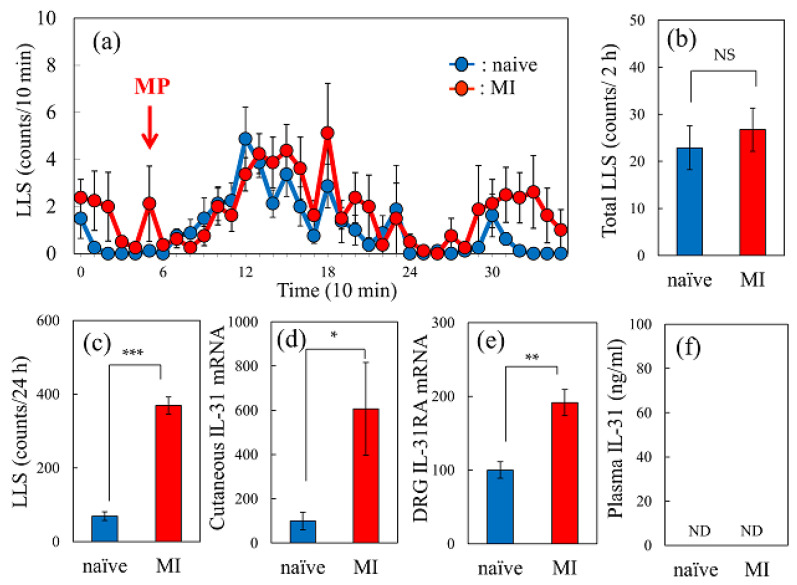
Comparative effect of morphine-induced LLS counts in naïve and mite-infested mice. (**a**) Time-course change of morphine-induced LLS counts in naïve and mite-infested (MI) mice. (**b**) Total LLS counts for 2 h after morphine administration. (**c**) Total LLS counts for 24 h before morphine administration. (**d**) Cutaneous IL-31RA expression before morphine administration. (**e**) DRG neuronal IL-31RA expression levels before morphine administration. (**f**) Serum IL-31 concentration in naïve and mite-infested (MI) mice. The red arrow indicates the morphine (3 mg/kg, subcutaneous) administration point. The blue line indicates the LLS counts in naïve mice; the red line indicates LLS counts in mite-infested (MI) mice. Each value represents the mean ± S.E. of six mice (total, 24 mice). NS, not significant. * *p* < 0.05, ** *p* < 0.01, and *** *p* < 0.001 compared with the corresponding values of naïve mice (Student’s *t*-test).

**Figure 7 ijms-24-16548-f007:**
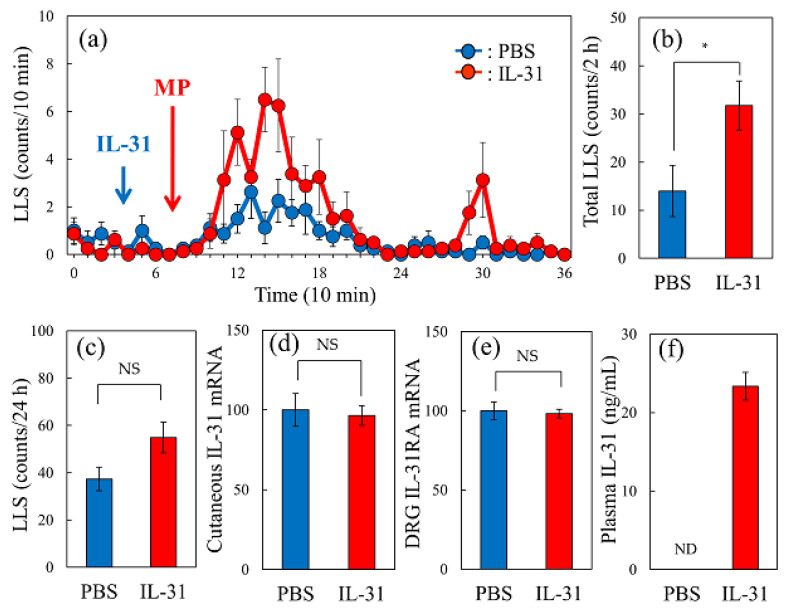
Effect of an intravenous injection of IL-31 on morphine-induced LLS counts in BALB/c mice. (**a**) Time-course change of vehicle (phosphate-buffered saline, PBS, 10 mL/kg, intravenous) or IL-31 (1 mg/kg, intravenous) on morphine-induced LLS counts in BALB/c mice. (**b**) Total LLS counts for 2 h after morphine administration. (**c**) Total LLS counts for 24 h before morphine administration. (**d**) Cutaneous IL-31RA expression levels before morphine administration. (**e**) DRG neuronal IL-31RA expression levels before morphine administration. (**f**) Serum IL-31 concentration 1 h after PBS or IL-31 administration. The red arrows indicate the morphine (3 mg/kg, subcutaneous) administration point. The blue arrows indicate the PBS or IL-31 (1 mg/kg, intravenous) administration point. The blue lines and columns indicate the values of the PBS-treated group, while the red lines and columns indicate the values of the IL-31-treated group. Each value represents the mean ± S.E. of six mice (total, 24 mice). NS, not significant. * *p* < 0.05 compared with the value of the PBS-treated group (Student’s *t*-test).

**Figure 8 ijms-24-16548-f008:**
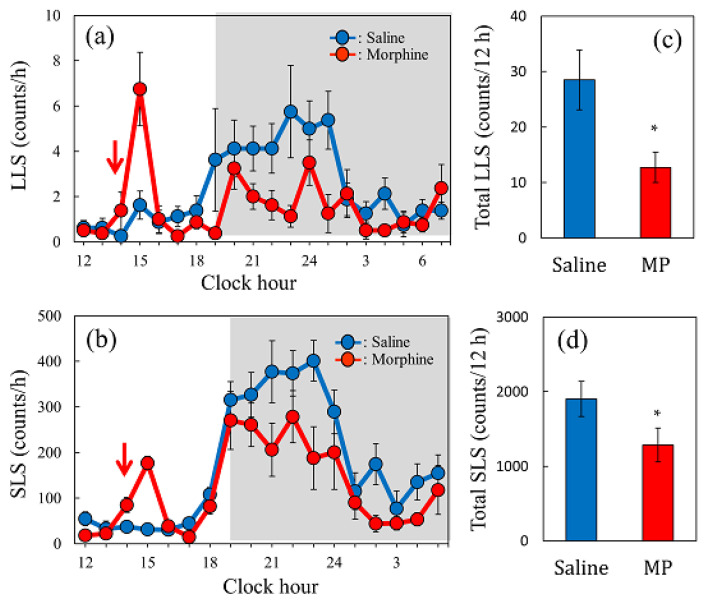
Effect of a single dose of morphine on spontaneous scratching behavior in mice. (**a**) Time-course change of LLS (itch-associated scratching behavior) after a single dose of morphine (5 mg/kg, subcutaneous) administration in mice. (**b**) Time-course change of SLS (hygienic behavior) after morphine administration in mice. Each value represents the mean ± S.E. of six mice. The red arrow indicates saline or morphine administration. The lateral axis indicates the clock hour, and the shaded area represents dark phase (7:00 p.m. to 7:00 a.m.). (**c**) Total LLS counts for 12 h from 6 h after morphine administration. (**d**) Total SLS counts for 12 h from 6 h after morphine administration. Each value represents the mean ± S.E. of six mice (total 12 mice). * *p* < 0.05 compared with each value of the saline-treated group (Student’s *t*-test).

**Figure 9 ijms-24-16548-f009:**
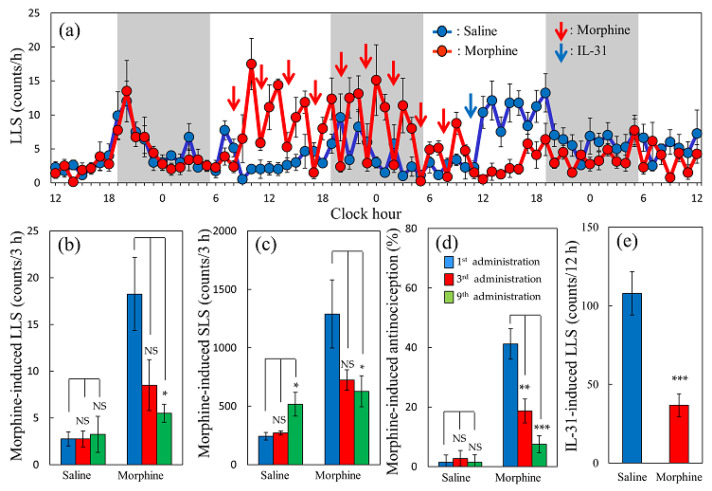
Effect of repeated administration of morphine on morphine-induced scratching counts and antinociceptive activity in mice. (**a**) Time-course changes of repeated administration on morphine (5 mg/kg, subcutaneous)-induced LLS (itch-associated scratching behavior) in mice. The red arrows indicate vehicle (saline, 10 mL/kg, subcutaneous) or morphine (5 mg/kg, subcutaneous, every 3 h for 27 h) administration, the lateral axis indicates the clock hour, and the shaded area represents dark phase (19:00 to 7:00). The blue line indicates the saline-administered group; the red line indicates morphine-administration group. The red arrows indicate the saline or morphine administration points. The blue arrow indicates vehicle (phosphate-buffered saline, PBS, 10 mL/kg, intravenous) or IL-31 (1 mg/kg, intravenous) administration point. (**b**) Total LLS (counts/3 h) of repeated administration of saline or morphine of the first, third, or ninth administration group. (**c**) Total SLS (counts/3 h) of repeated administration of morphine of the first, third, or ninth administration group. (**d**) Antinociception index of repeated administration of saline or morphine of the first, third, or ninth administration group. The blue columns indicate the response of the first morphine-administration group; the red columns indicate the third morphine-administration group; and the green columns indicate the ninth morphine-administration group. The results are presented as the mean ± S.E. of 6 mice. * *p* < 0.05, ** *p* < 0.01, and *** *p* < 0.001 compared with those of the values of first morphine-administration group. Results are presented as the mean ± S.E. of 6 mice (total, 12 mice) (Dunnett test). (**e**) Effect of repeated administration of morphine on IL-31 (1 mg/kg, intravenous)-induced LLS counts in mice. Each value represents the mean ± S.E. of six mice (total, 12 mice). *** *p* < 0.001 compared with the PBS-administration group (Student’s *t*-test).

**Figure 10 ijms-24-16548-f010:**
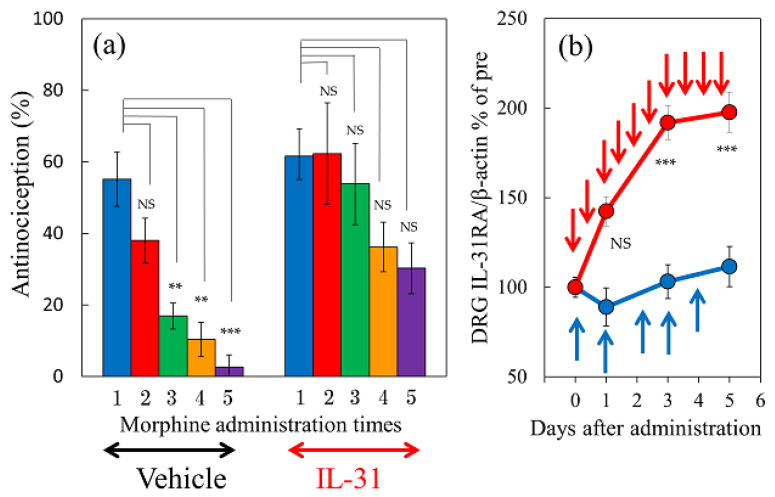
Effects of combined administration of IL-31 and morphine on morphine-induced antinociceptive tolerance and DRG neuronal IL-31RA mRNA expression in mice. (**a**) Time-course change of combined administration of morphine (10 mg/kg, subcutaneous, once a day for 5 days) and vehicle (phosphate-buffered saline, PBS 10 mL/kg, intraperitoneal, every 12 h for 5 days) or IL-31 50 μg/kg, intraperitoneal, every 12 h for 5 days) on morphine-induced antinociceptive tolerance. Each column represents the mean ± S.E. of six mice (total, 12 mice). NS, not significant, ** *p* < 0.01, and *** *p* < 0.001 compared with the values of first morphine-administration of each PBS or IL-31 treated group (Dunnet-test). (**b**) Time-course change of combined administration of morphine and IL-31 on DRG neuronal IL-31RA expression. The blue line indicates the PBS administered group; the red line indicates IL-31 administration group. Each column represents the mean ± S.E. of six mice (total, 48 mice). NS, not significant, *** *p* < 0.001 compared with the corresponding values of each PBS + morphine-treated group (Student’s *t*-test with Bonferroni correction). The time-course change of repeated administration of PBS and morphine (10 mg/kg, subcutaneous, once a day for 5 days) did not significantly change DRG neuronal IL-31RA expression compared to that before morphine administration (Figure 10b, blue line). However, combined repeated administration of IL-31 (50 μg/kg, intraperitoneal, every 12 h for 5 days) and morphine (10 mg/kg, subcutaneous, once a day for 5 days) significantly increased DRG neuronal IL-31RA expression compared with the morphine-treated group (Figure 10b, red line).

**Figure 11 ijms-24-16548-f011:**
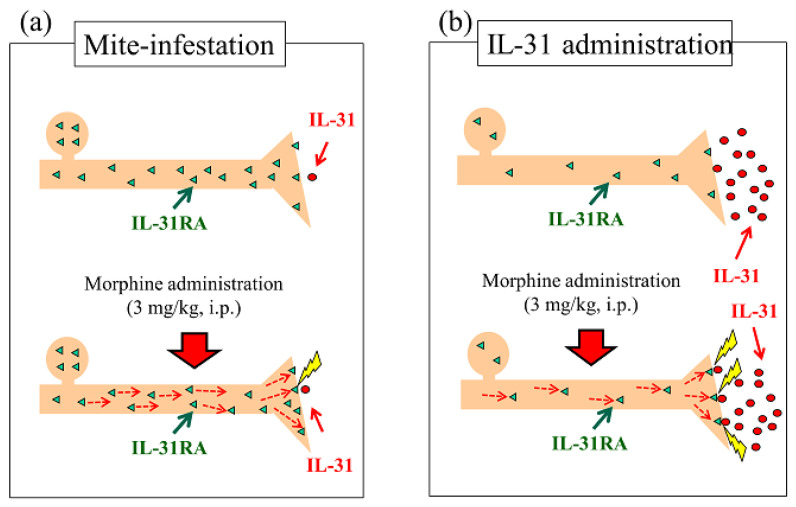
Schematic of the association of intraneural IL-31RA and blood IL-31 concentration on naïve or mite-infested BALB/c mice. (**a**) Mite-infested mice were the result of co-housing mice with skin-lesioned NC/Nga mice. Mite infestation significantly increased DRG neuronal IL-31RA expression and not the blood concentration of IL-31 (upside). The morphine-induced LLS counts did not change in mite-infested mice (downside). (**b**) IL-31 treatment (1 mg/kg, intravenous injection) 1 h before morphine administration; IL-31 administration does not increase IL-31RA expression in the skin and neuronal DRG; however, it markedly increases the blood concentration of IL-31 (upside). Morphine-induced LLS counts were significantly enhanced compared with those of the saline-treated group. These results suggest that morphine may promote the exocytosis of IL-31RA in peripheral sensory nerve endings.

**Figure 12 ijms-24-16548-f012:**
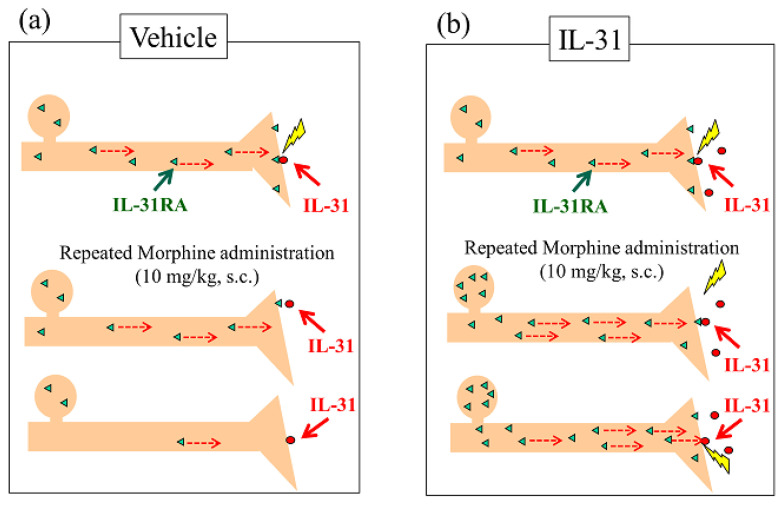
Schematic of intraneural IL-31RA content upon repeated administration of morphine and the effect of combination treatment of IL-31 in mice. (**a**) Effect of repeated administration of morphine. IL-31RA content in the presynaptic nerve gradually decreases upon repeated morphine administrations (middle and downside). LLS counts and antinociceptive activity also decrease, and analgesic tolerance is formed. (**b**) Effect of combination treatment of repeated administration of morphine and IL-31. IL-31RA content in the presynaptic nerve gradually decreases upon repeated morphine administrations; however, IL-31RA decrease in the presynaptic nerve is controlled by the increase in IL-31RA expression by IL-31, and the analgesic tolerance formation relaxes (middle and downside).

## Data Availability

Data are contained within the article.

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
