# Peer review of "Experimental Study: Interleukin-31 Augments Morphine-Induced Antinociceptive Activity and Suppress Tolerance Development in Mice"

_ijms, 2023, doi:10.3390/ijms242216548_

Round 1

Reviewer 1 Report

Comments and Suggestions for Authors

While the manuscript is scientifically sound, the abstract should be improved, it needs to be revised to succinctly consolidate the results and draw a cohesive conclusion.

Unit of Antinociceptive Activity:

The unit of antinociceptive activity in mice is not indicated in Figures 2a, 3a, 3b, and 4a.

Age Discrepancy Among Mice:

Can you provide a clear rationale regarding the age difference between the skin-lesioned-NC/Nga mice (aged 10–13 weeks) and BALB/c and C57BL/6 mice (aged 6–8 weeks).

Harvesting Neuronal Cell Bodies:

Provide a precise description of the methodology employed for harvesting neuronal cell bodies and clarify the statement "neuronal cell bodies from the shoulders and backs of BALB/c and C57BL/6 mice at each point" in the QPCT methods.

Reference Citations:

The references should be augmented with more recent citations to ensure that the body of work remains up-to-date and relevant.

Comments on the Quality of English Language

Minor errors are to be corrected.

Author Response

Thank you for your helpful advice for our manuscript. We have modified the original draft according to your suggestions as follows.

  1. Unit of Antinociceptive Activity: The unit of antinociceptive activity in mice is not indicated in Figures 2a, 3a, 3b, and 4a.

Response 1.  Thank you for your thorough review. We apologize for this error; we have added the unit % for antinociceptive activities in the revised text (Figure 2a, 3a, 3b and 4a).

  1. Age Discrepancy Among Mice: Can you provide a clear rationale regarding the age difference between the skin-lesioned-NC/Nga mice (aged 10–13 weeks) and BALB/c and C57BL/6 mice (aged 6–8 weeks).

Respons 2.  The NC/Nga mice was not used to measure antinociceptive activity and scratching counts and instead cohoused with BALB/c mice continuously to increase DRG neuronal IL-31RA levels. As the mite-infection power was good in NC/Nga mice more than 10 weeks of age, we used an age in 10-13 week mice.

We have added the sentence as follows:

Page 16, lines 34–35.

We used the NC/Nga mice to increase IL-31RA levels in BALB/c mice.

  1. Harvesting Neuronal Cell Bodies: Provide a precise description of the methodology employed for harvesting neuronal cell bodies and clarify the statement "neuronal cell bodies from the shoulders and backs of BALB/c and C57BL/6 mice at each point" in the QPCT methods.

Reference Citations: The references should be augmented with more recent citations to ensure that the

Respons 3. It is the information of the anatomic textbook not references.

We have added the reference as follows: Page 21, lines 35–36.

  1. Hannsjörg Schröder, Natasha Moser, Stefan Huggenberger. Neuroanatomy of the Mouse; Springer Nature Switzerland AG 2020. pp: 59-78.

Reviewer 2 Report

Comments and Suggestions for Authors

The authors have studied the effects of IL-31 on morphine tolerance in the induction of antinociception and scratching, effects that appear to be correlated. The results appear to be well performed are clearly presented. The positive effects of IL31 with respect to tolerance are interesting. While the work has merit, the manuscript needs editing. First, the overall presentation and discussion is too long and in parts difficult to read. Many hypotheses are discussed that do not directly relate to the presented results. Here are a few comments:

1. What is the potential clinical utility of IL31? What adverse effects can one expect from IL31 in the clinic? It would help to address these issues. Could this be a useful strategy during long-term opioid pain therapies?

2.  The Introduction is too long, distracting from the principal goals of the study.

3. Discussion. Delete general reviews of opioid pharmacology, for example: “Based on pharmacological and molecular cloning studies, distinct-, m-, d-, and k- opioid receptors have been identified. Several, researchers have studied the analgesic activity of opioids based on these receptors; however, the precise mechanism remains unknown.”

These sentences are difficult to read and need editing: “Recently, we reported that pretreatment with IL-31 cause alloknesis (LLS) and antinociceptive activity simultaneously in mice [12]. In contrast, morphine-induced LLS and antinociceptive effects were closely correlated in wild-type mice.”   … “Morphine-induced antinociception enhanced by pretreatment with IL-31, increasing doses of morphine or IL-31.”

What is the evidence for IL-31RA exocytosis presumably as a mechanism of downregulation?  “Our hypothesis links IL-31 and morphine via a potential site that promotes IL-31RA exocytosis, thus reinforcing the antinociceptive activity of morphine.”

Page 14.  Is this paragraph needed?  “Analgesic sensitivity to opioids, such as morphine ….”.  The manuscript would benefit from thoroughly editing for brevity to highlight the salient results in a readily readable format.

Comments on the Quality of English Language

Some editing is needed to improve readability, including shortening.

Author Response

We thank you for your thoughtful suggestions and insights.

  1. What is the potential clinical utility of IL31? What adverse effects can one expect from IL31 in the clinic? It would help to address these issues. Could this be a useful strategy during long-term opioid pain therapies?

Response 1: To date, to the best of our knowledge, no clinical studies on the analgesic action of IL-31 have been conducted. However, phenomena such as individual of analgesic sensitivity or antinociceptive tolerance be up by clinical practice of morphine so that the possibility improved by IL-31 is high as even a mouse is up.

  1. The Introduction is too long, distracting from the principal goals of the study.

Response 2: According to the reviewer’s suggestion, we have deleted the following portion from the revised text.

via two possible mechanisms. The first is involved in the signal transduction of opioid receptors, including receptor down-regulation, functional decoupling from G-proteins, and β-arrestin recruitment [4,5]. The second is characterized by alterations in primary drug-sensitive system adaptation [6] and glial cell activation [7], thus inhibiting the analgesic effect of morphine. Tolerance to opioid analgesics is a major concern associated with these drugs; however, an effective solution is still needed.

3-1. Discussion. Delete general reviews of opioid pharmacology, for example: “Based on pharmacological and molecular cloning studies, distinct-, m-, d-, and k- opioid receptors have been identified. Several, researchers have studied the analgesic activity of opioids based on these receptors; however, the precise mechanism remains unknown.”

Response 3-1: According to the reviewer’s suggestion, we have deleted the above sentence from the revised text.

3-2. These sentences are difficult to read and need editing: “Recently, we reported that pretreatment with IL-31 cause alloknesis (LLS) and antinociceptive activity simultaneously in mice [12]. In contrast, morphine-induced LLS and antinociceptive effects were closely correlated in wild-type mice.”   … “Morphine-induced antinociception enhanced by pretreatment with IL-31, increasing doses of morphine or IL-31.”

Response 3-2: According to the reviewer’s suggestion, we have revised following sentence from the revised text.

Page 13, lines 6–10.

Recently, we reported that pretreatment with IL-31 cause alloknesis (LLS) and antinociceptive activity simultaneously in mice [12]. In contrast, morphine-induced LLS and antinociceptive effects were closely correlated in wild-type mice. Repeated pre-administration of IL-31 marginally increased the morphine-induced antinociceptive effects in the hot-plate test.

What is the evidence for IL-31RA exocytosis presumably as a mechanism of downregulation?  “Our hypothesis links IL-31 and morphine via a potential site that promotes IL-31RA exocytosis, thus reinforcing the antinociceptive activity of morphine.”

Response 3-3: The site of action of morphine is the pre-synaptic sensory nervous system. The antinociceptive activity of morphine was partially inhibited in IL-31RAKI mice. IL-31RA effect disappears on frequent administering morphine.

We have added the sentence as follows: Page 13, lines 39-43.

The site of action of morphine is the pre-synaptic sensory nervous system. The antinociceptive activity of morphine was partially inhibited in IL-31RAKI mice. IL-31RA effect disappears on frequent administering morphine. Repeated morphine administration was no effect on DRG neuronal IL-31RA expression levels. From these results …

Page 14.  Is this paragraph needed?  “Analgesic sensitivity to opioids, such as morphine ….”.  The manuscript would benefit from thoroughly editing for brevity to highlight the salient results in a readily readable format.

Response 3-4: As per to the reviewer’s suggestion, we have deleted the above sentence from the revised text.
